# A Duplicated Copy of the Meiotic Gene *ZIP4* Preserves up to 50% Pollen Viability and Grain Number in Polyploid Wheat

**DOI:** 10.3390/biology10040290

**Published:** 2021-04-02

**Authors:** Abdul Kader Alabdullah, Graham Moore, Azahara C. Martín

**Affiliations:** Crop Genetics Department, John Innes Centre, Colney, Norwich NR4 7UH, UK; Abdul-Kader.Alabdullah@jic.ac.uk (A.K.A.); Azahara-C.Martin@jic.ac.uk (A.C.M.)

**Keywords:** wheat, polyploidy, meiosis, *ZIP4*, *Ph1* gene, pollen analysis, fertility

## Abstract

**Simple Summary:**

On wheat polyploidisation, the major meiotic gene *ZIP4*, duplicated and diverged, such that tetraploid and hexaploid wheat each carry three and four copies of *ZIP4*, respectively. Surprisingly, this study demonstrates that, in hexaploid wheat, despite the presence of the other three *ZIP4* copies, the duplicated *ZIP4* copy is required to prevent major abnormalities during meiosis. Although there is greater disruption of subsequent male rather than female fertility, the duplicated *ZIP4* copy preserves up to 50% of the grain number. High wheat fertility is important since it is consumed by over 4.5 billion people on the planet, of whom 2.5 billion are dependent on it. This study highlights the potentially extraordinary value of the wheat *ZIP4* duplication, mandating further studies to unravel the complexity of the *ZIP4* phenotype in this global crop.

**Abstract:**

Although most flowering plants are polyploid, little is known of how the meiotic process evolves after polyploidisation to stabilise and preserve fertility. On wheat polyploidisation, the major meiotic gene *ZIP4* on chromosome 3B duplicated onto 5B and diverged (*TaZIP4-B2*). *TaZIP4-B2* was recently shown to promote homologous pairing, synapsis and crossover, and suppress homoeologous crossover. We therefore suspected that these meiotic stabilising effects could be important for preserving wheat fertility. A CRISPR *Tazip4-B2* mutant was exploited to assess the contribution of the 5B duplicated *ZIP4* copy in maintaining pollen viability and grain setting. Analysis demonstrated abnormalities in 56% of meiocytes in the *Tazip4-B2* mutant, with micronuclei in 50% of tetrads, reduced size in 48% of pollen grains and a near 50% reduction in grain number. Further studies showed that most of the reduced grain number occurred when *Tazip4-B2* mutant plants were pollinated with the less viable *Tazip4-B2* mutant pollen rather than with wild type pollen, suggesting that the stabilising effect of *TaZIP4-B2* on meiosis has a greater consequence in subsequent male, rather than female gametogenesis. These studies reveal the extraordinary value of the wheat chromosome 5B *TaZIP4-B2* duplication to agriculture and human nutrition. Future studies should further investigate the role of *TaZIP4-B2* on female fertility and assess whether different *TaZIP4-B2* alleles exhibit variable effects on meiotic stabilisation and/or resistance to temperature change.

## 1. Introduction

Polyploidy occurs in a wide range of species, including fish, flatworms, shrimp, amphibians, flowering plants, wine and brewing yeast [1,2,3,4]. The molecular mechanisms responsible for meiotic polyploidisation and diploid behaviour are important for ensuring correct chromosome segregation of multiple related chromosomes, production of balanced gametes and hence preservation of fertility. It is surprising that these mechanisms have not been more widely investigated, given their potentially enormous value to humankind [4]. 

Polyploids can either arise via multiplication of a basic set of chromosomes (termed an autopolyploid, such as potato) or by combining related but not completely identical genomes (termed an allopolyploid, such as hexaploid (bread) wheat). Plant polyploidisation is often associated with extensive chromosomal rearrangements and changes in gene content and expression [3,5,6,7]. Yet, analysis of the recently sequenced hexaploid wheat (*Triticum aestivum* L.) genome and wheat RNA seq datasets from over 1000 tissues (including meiocytes), did not reveal extensive gene loss or changes in expression between related (homoeologous) chromosomes following polyploidisation [8]. Even meiotic genes do not appear to have suffered gene loss, exhibiting mostly balanced expression between copies on related chromosomes (homoeologues) [9]. Thus, hexaploid wheat appears to have suffered less extensive rearrangement.

It was previously accepted that a locus arising on chromosome 5B during wheat polyploidisation, was responsible for stabilising the wheat genome during meiosis, hence maintaining fertility. This was based on earlier cytogenetic studies of hexaploid wheat lines lacking the whole of chromosome 5B, which when crossed with wild relatives, exhibited homoeologous crossover between wheat and wild relative chromosomes at metaphase I in the resulting hybrids [10,11]. It was unclear at the time which and how many of these genes needed to be lost to produce the phenotype. However, it was recognised from these early studies that suppression of homoeologous crossover was important for stabilising the wheat genome and maintaining its fertility. In 1971, a study coined the term ‘pairing homoeologous’ (*Ph1*) for this ‘critical locus’ on 5B, responsible for suppressing the homoeologous crossover observed in wheat-wild relative hybrids [12]. Loss of *Ph1* (or of the whole 5B chromosome, as in these studies) allowed homoeologous crossover to take place. The term ‘pairing’ was used synonymously with crossover observed at metaphase I at this time. ‘*Ph1*′ became the accepted term to describe the locus responsible for the homoeologous crossover suppression phenotype.

Sears [13] identified a mutant (named *ph1b*) carrying a deletion of part of chromosome 5B (now known to be 59.3 Mb in size, encompassing some 1187 genes [14]). When the Sears *ph1b* mutant was crossed with wild relatives to form hybrids, crossover between homoeologues was subsequently observed during meiosis in these hybrids. Later, Roberts et al. [15] observed that mutants carrying deletions in the long arm of chromosome 5B could be separated into 2 groups by scoring the configurations at metaphase I during meiosis in the mutants. Univalents, rod bivalents and multivalents were present in over 50% of meiocytes at metaphase I in the Sears *ph1b* mutant (and also some of the 5B deletion mutants), while the wild type (WT) wheat and the remaining 5B deletion mutants exhibited mainly bivalents at metaphase I in all their meiocytes. Thus, the presence of meiotic abnormalities in over 50% of meiocytes could separate the 5B deletion mutants into two distinct groups [15]. Presence of multivalents suggested that the initial alignment of chromosomes (now termed ‘pairing’) and intimate pairing (now termed ‘synapsis’) of chromosomes was disrupted.

Recently the two phenotypes (suppression of homoeologous crossover (*Ph1*) in wheat-wild relative hybrids, and the presence of meiotic abnormalities in 50% of meiocytes in the 5B deletion mutants themselves) have been defined using a series of 5B deletions to a 0.5 Mb region of chromosome 5B containing a copy of the major meiotic gene *ZIP4* (*TaZIP4-B2*) [14,16,17,18,19]. Genome analysis revealed that hexaploid wheat possessed a further *ZIP4* gene on each of the group 3 chromosomes. Analysis by the International Wheat Genome Sequencing Consortium [20] confirmed that on wheat polyploidisation, *TaZIP4-B2* was derived from *ZIP4 3B* (*TaZIP4-B1)* through a trans-duplication event. *ZIP4* is a protein containing tetratricopeptide repeats (TPRs), which can assemble protein complexes promoting homologous crossover in plants [21,22]. In the wheat CRISPR *Tazip4-B2* deletion mutant, 50% of meiocytes exhibit meiotic abnormalities (the phenotype reported by Roberts et al. [15]) [19], demonstrating that *TaZIP4-B2* promotes homologous pairing, synapsis and crossover. However, when the CRISPR *Tazip4-B2* deletion mutant is crossed with a wild relative to form a hybrid, homoeologous crossover also takes place [19], implying a role for *TaZIP4-B2* in suppression of homoeologous crossover. Thus, the duplication of *ZIP4* on wheat polyploidisation led to an adaptation during meiosis I, preventing meiotic disruption by promoting homologous pairing, synapsis and crossover, and suppressing homoeologous crossover.

Following meiosis I, the reductional division during wheat male meiosis (meiosis II), leads to the formation of tetrads each containing four microspores, which degenerate to release individual uninucleate microspores. Each microspore undergoes mitotic divisions to produce a mature trinucleate pollen grain, with one vegetative nucleus and two generative nuclei or sperm cells. Reductional division in wheat female meiosis results in a T-shaped tetrad, containing 4 megaspores. Only one of the megaspores develops into an embryo sac, with the remaining 3 megaspores degenerating. Hence, whilst all four products of meiosis survive on the male side, only one survives on the female side. Meiosis is an essential process for the formation of gametes. Thus, meiotic abnormalities or genetic disruptions are likely to result in reduced fertility. Meiotic abnormalities on the male side may be associated with variable sized and/or inviable pollen grains, and on the female side, with a partial reduction in grain number or complete sterility [23,24,25].

In the polyploid literature, it is often stated that meiotic adaptation is important for polyploid fertility. However, it has not previously been possible to determine the effect of an actual meiotic adaptation. The availability of a CRISPR deletion mutant for the duplicated *TaZIP4-B2* copy allows us to assess the effect of this meiotic adaptation on the correct segregation of chromosomes, effective production of balanced gametes, and hence preservation of pollen viability and grain number in this major global crop. As part of this assessment, a pollen profiling method has been developed and exploited to compare pollen profiles of different mutants in hexaploid (and tetraploid) wheat.

## 2. Materials and Methods

### 2.1. Plant Material

Three different *Tazip4-B2* mutants were used in this study: (1) *ph1b*: a hexaploid wheat *T. aestivum* cv. Chinese Spring mutant [13]; (2) CRISPR *Tazip4-B2*: a hexaploid wheat *T. aestivum* cv. Fielder mutant with a CRISPR-induced deletion of 38 amino acids (A^104^ to E^141^) from the TaZIP4-B2 protein [19]; (3) *ph1c***:** a tetraploid wheat *T. turgidum* subsp. *Durum* cv. Senatore Cappelli mutant [26].

### 2.2. Pollen Profiling

Ten plants for each genotype were grown in a controlled environment room (CER) at 20 °C (day) and 15 °C (night) (16-hr photoperiod, 70% humidity). Mature yellow anthers were collected just before shedding pollen, from five main florets at the middle portion of the first spike of each plant. Each of three anthers from the same floret were placed in an Eppendorf containing 0.5 mL 70% ethanol and stored at 4 °C. Pollen grains were released from anthers by sonication and the sonicated pollen samples filtered through 200 µm sieves using 100 mL Coulter Isoton II diluent (Beckman Coulter). Size and number of filtered pollen grains were measured using a Coulter counter (Multisizer 4e, Beckman Coulter Inc., Brea, CA, USA), fitted with a 200 µm aperture tube, with Isoton II diluent (using the following settings: Control mode: volumetric; Analytic volume: 2000 µL; Electrolyte volume: 100 mL; Size bins = 400 from 4 μm to 120 μm; Current: 1600 µA; Stirring speed: 20 CW). For each sample, the measured pollen number distribution over size bins was exported into a csv file, then an R script (Text S1) used to extract and calculate plot-differential pollen size distribution and pollen number per anther from the raw data files for each genotype.

### 2.3. Pollen Viability

Pollen viability was assessed using Alexander stain [27]. Fresh wheat pollen grains from three anthers were shed on a droplet of Alexander stain placed on a microscope slide and images taken for scoring. Three biological replicates, each with >1000 pollen grains were analysed for each genotype.

### 2.4. Grain Number Per Spike Assessment

In the CER, plants were grown at 20 °C (day) and 15 °C (night) (16-h photoperiod, 70% humidity). In the glasshouse, 11–15 plants from each genotype were grown at 22 °C (day) and 17 °C (night) (16-h photoperiod, 70% humidity). In both experiments, the first three bagged spikes from each plant were harvested when fully dried, and threshed separately after counting spikelet number. The number of grains per spike was then measured using the MARVIN grain analyser (GTA Sensorik GmbH, Neubrandenburg, Germany). Grain number per spike ((actual grain number per spike/expected grain number per spike) × 100) was normalised in order to eliminate the effect of different number of spikelets per spike on grain number. Expected grain number was calculated by multiplying number of spikelets by three, considering that each spikelet has three main fertile florets.

### 2.5. Female Sterility Assessment

Female sterility was assessed through the emasculation/pollination method, using the CRISPR *Tazip4-B2* mutant as both pollen donor and recipient. For each pollination experiment, at least twelve spikes were emasculated at the heading stage when the spike had fully emerged from the flag leaf and anthers were still green with a tight stigma. Spikelets located at the tip and base of the spike, and florets in the centre of each spikelet were removed before emasculation. Emasculated spikes were covered with crossing bags. Receptive and mature stigma were pollinated using fresh pollen grains collected from fully mature anthers. All emasculations and pollinations were undertaken in the morning. Grain number per spike was normalised by dividing the number of grains by the number of pollenated florets per spike.

### 2.6. Seed Germination Rate

Seeds resulting from the emasculation/pollination experiment were disinfected with 5% Sodium Hypochlorite, washed with distilled water, placed on wet filter paper in Petri dishes (9 cm diameter), wrapped with aluminium foil and placed in a growth chamber for 10 days (22 °C). Each Petri dish represented a replicate containing 15 seeds originating from the same spike. Five replicates were used for each treatment. Seeds were considered to have germinated after radicle emergence. The germination percentage was calculated as (number of seeds germinated/total seeds) × 100.

### 2.7. Meiotic Analysis

Anthers were collected and fixed as previously described [28]. Cytological analysis of Pollen Mother Cells using the Feulgen technique was performed as previously described [29]. FISH preparations of repetitive sequences, 4P6 and pTa71, and FISH analysis of anthers fixed at the tetrad stage were as described previously [30]. Digoxigenin-labelled probes were detected with anti-digoxigenin-fluorescein Fab fragments (Sigma), and Biotin-labelled probes were detected with Streptavidin-Cy5 (Thermo Fisher Scientific, Waltham, MA, USA).

### 2.8. Image Processing

Pollen grains and Pollen Mother Cells stained by the Feulgen technique were imaged using a LEICA DM2000 microscope (Leica Microsystems, http://www.leica-microsystems.com/, accessed on 31 March 2021), equipped with a Leica DFC450 camera and controlled by LAS v4.4 system software (Leica Biosystems, Wetzlar, Germany). Tetrads labelled by FISH were imaged using a Leica DM5500B microscope equipped with a Hamamatsu ORCA-FLASH4.0 camera and controlled by Leica LAS X software v2.0. Z-stacks were processed using the 561-deconvolution module of the Leica LAS X Software package. Images were processed using Adobe Photoshop CS5 (Adobe Systems Incorporated, San Jose, CA, USA) extended version 12.0 × 64.

### 2.9. TaZIP4 Protein Sequence Analysis

DNA, CDS and protein sequences of the four *TaZIP4* homologues were retrieved from the *Ensembl Plants* database for *Triticum aestivum* (IWGSC v1.1 gene annotation [20]). Multiple sequence alignments of coding sequences (CDS) and protein sequences of *TaZIP4-A1* (TraesCS3A02G401700.3), *TaZIP4-B1* (TraesCS3B02G434600.2), *TaZIP4-B2* (TraesCS5B02G255100.1), *TaZIP4-D1* (TraesCS3D02G396500.2) and the mutant CRISPR *Tazip4-B2* [19], were performed using the Clustal X programme (version 2; [31,32]). Programmes used for protein analysis are described below.

## 3. Results

### 3.1. Divergence and the CRISPR Deletion Occur within the TaZIP4-B2 TPR Domain

The InterProScan [33] and PFAM programmes identified a single highly conserved SPO22 domain (PF08631) within the EBI database ZIP4s. This SPO22 domain was composed of tetratricopeptide repeats (TPRs) of 34 amino acids. A second SPO22 domain of low significance was observed in tandem with the highly conserved SPO22 domain in many ZIP4s. Only PFAM classified this second SPO22 domain as being significant for a limited number of these ZIP4s. *ZIP4* function is dependent on these SPO22 TPR-containing domains, due to their involvement in assembling protein complexes [34,35]. The annotation programmes enabled us to assess whether the divergence of *TaZIP4-B2* from its chromosome group 3 homoeologues (*TaZIP4-A1*, *TaZIP4-B1* and *TaZIP4-D1*) occurred within the SPO22 domain. Similarly, we used the annotation programmes to determine the site of the in-frame 38 amino acid CRISPR deletion of *Tazip4-B2* [19] relative to the SPO22 domain. Multiple sequence alignments showed that TaZIP4-B2 was quite divergent from the other group 3 homoeologues (Figure 1a). The percentage of identity between *TaZIP4-B2* and the other homoeologues did not exceed 85.8% in coding sequences (CDS) and 92.2% in protein sequences, whereas the inter-identity of *TaZIP4-A1*, *TaZIP4-B1* and *TaZIP4-D1* ranged from 94.9–96.3% for CDS sequences and from 96.8–97.5% for protein sequences (Appendix A). The InterProScan and PFAM programmes identified the highly conserved SPO22 domain within all the wheat ZIP4s, with PFAM identifying a second SPO22 domain in tandem (Figure 1b). TPRpred [36] identified 12 TPRs within wheat ZIP4-B1 (Figure 1c), showing that up to half of total ZIP4-B1 protein consisted of TPRs. However, the region of TaZIP4-B2 corresponding to the 3rd TPR of ZIP4-B1 within the highly conserved SPO22 domain, was no longer identified as a TPR by TPRpred. Thus, within wheat TaZIP4-B2, only 11 TPRs were identified. The 2nd and 4th TPRs of TaZIP4-B2 and TaZIP4-B1 also exhibited some divergence with respect to each other. As a result of this divergence, the MARCOIL programme [36,37] suggested an altered conformation within the conserved SPO22 domain of TaZIP4-B2 compared to the domains of TaZIP4-B1 and other *ZIP4* homoeologues (Figure 1d). Thus, duplication of *TaZIP4-B2* from *TaZIP4-B1* led to TPR divergence (especially the 3rd TPR (Appendix A)), giving rise to associated changes in protein conformation. This predicts that *TaZIP4-B2* function may be altered with respect to that of its chromosome group 3 homoeologues. The 38 amino acid in-frame CRISPR deletion (within the CRISPR *Tazip4-B2*) covered the 1st TPR, indicating that the deletion did indeed affect the SPO22 domain and correlated with complete loss of the *TaZIP4-B2* phenotype [19].

### 3.2. Effect of the TaZIP4-B2 Deletion on Meiotic and Tetrad Stages

Average meiotic scores from *Tazip4-B2* mutants at metaphase I were reported previously [19,38]. However, the present study required meiotic scores from individual meiocytes from *Tazip4-B2* mutants at metaphase I, in order to relate meiotic abnormalities observed during metaphase I with those observed at subsequent stages. Both CRISPR *Tazip4-B2* [19] and *ph1b* [13] mutants were exploited. The *ph1b* mutant carries a 59.3 Mb deletion encompassing some 1187 genes, including *TaZIP4-B2* [14]. Meiotic scores from individual meiocytes at metaphase I from CRISPR *Tazip4-B2* [19], *ph1b* [38] and their respective wild type plants are provided in Table 1 and Appendix A and visualised in Figure 2. Examples of meiotic configurations of the CRISPR *Tazip4-B2* mutant and wild type (WT Fielder) plants at metaphase I are provided in Figure 3a,b. More than half of the scored meiocytes in the CRISPR *Tazip4-B2* and *ph1b* mutants had meiotic abnormalities (Figure 2). Overall, univalents and/or multivalents were observed in 56% of both the CRISPR *Tazip4-B2* and *ph1b* mutant meiocytes. Univalents were present in 49% and 43% of meiocytes at metaphase I (average per meiocyte 1.16 and 0.8) for the CRISPR *Tazip4-B2* and *ph1b* mutants, respectively. Multivalents were present in 32% and 43% (average 0.39 and 0.53 per meiocyte) of the CRISPR *Tazip4-B2* and *ph1b* mutant meiocytes, respectively. The slight excess of multivalents and lack of univalents in the *ph1b* mutant compared to the CRISPR *Tazip4-B2* mutant may be simply due to accumulation of the extensive rearrangements observed and reported in this mutant [14], which can form multivalents at metaphase I (Table 1 and Appendix A). The excess of multivalents in the *ph1b* mutant could also be explained by additional unknown deleted genes within the 59.3 Mb 5B deletion, but the issue cannot be resolved by just scoring for the presence or absence of multivalents at metaphase I, as scoring multivalents alone would fail to distinguish between different deletion mutants covering variable lengths of the long arm of 5B [15]. Thus, the deletion of *TaZIP4-B2* leads to nearly half of meiocytes possessing univalents as a result of pairing and crossover failure, and a third of meiocytes possessing multivalents as a result of incorrect pairing and crossover.

Meiotic aberrations at metaphase I (univalents and multivalents) can lead to imbalanced chromosomal segregation at anaphase I, with subsequent disruption to the post-meiotic process. Giorgi had already reported the presence of lagging chromosomes at anaphase I and micronuclei at the tetrad stage in the tetraploid *Ph1* deletion mutant (*ph1c*) carrying a large chromosomal deletion covering the *TaZIP4-B2* region [26]. Therefore, the stages following metaphase I were studied in both the WT Fielder and the CRISPR *Tazip4-B2* mutant. In WT Fielder, homologous chromosomes (homologues) appear connected to each other by one or mostly several chiasmata (Figure 3a), with only an occasional univalent being present during metaphase I. Each homologue separates to a different pole of the nucleus during anaphase I, resulting in equal separation of homologues (Figure 3c,e). After the second meiotic division, tetrads with four balanced gametes each are formed (Figure 3g). In the CRISPR *Tazip4-B2* mutant, univalents, multivalents and a global reduction in the number of chiasmata were observed at metaphase I (Figure 3b), as previously reported [19]. Although unbalanced segregation of chromosomes would be expected during anaphase I as a consequence of disrupted crossover distribution, observed disruptions were greater than expected, with regular presence of lagging chromosomes, split sister chromatids and chromosome fragmentation (Figure 3d,f). The high number of micronuclei (MN) observed in tetrads, the final product of meiosis, was the most surprising result (Figure 3h,j). It is not unusual to find an occasional MN in wheat. Indeed, some were found in the WT Fielder analysed in this study (less than 5% of tetrads), probably due to an occasional univalent observed at metaphase I. However, in the CRISPR *Tazip4-B2* mutant, it was striking that more than 50% of tetrads showed at least one MN (Figure 3i); one, two and less frequently three MN per tetrad were detected. Fluorescent in situ hybridisation (FISH) was performed on tetrads from both the WT Fielder and CRISPR *Tazip4-B2* mutant, using the repetitive probes 4P6 [39] and pTa71 [40], in order to assess the level of mis-segregation and to ascertain whether specific chromosomes were involved in MN formation. Probe 4P6 labels seven interstitial sites on D genome metaphase I chromosomes, while pTa71 labels the NOR (Nucleolar Organiser Region) on the 1BS, 6BS and 5DS metaphase I chromosomes. A 4P6 signal was observed in 23.8% MNs, confirming a D genome chromosome origin, and a pTa71 signal in 17.5% MNs, indicating that some chromosomes were carrying a NOR. This suggests that MN formation did not result from a single specific pair of homologues being univalent at metaphase I, but rather from different pairs of homologues being univalent in individual meiocytes. Morrison [41] observed that univalents at metaphase I lagged at anaphase I, and then formed MN at the dyad stage. Such MNs were then maintained until the tetrad stage, when they were lost with the separation of the four microspores. Morrison [41] also observed a direct correlation between numbers of univalents at metaphase I and percentage of tetrads with MN. As such, our observations are consistent with those of Morrison [41], in that 56% of *Tazip4-B2* mutant meiocytes exhibited abnormalities at metaphase I, while 50% of tetrads subsequently possessed MN.

### 3.3. Effect of the Tazip4-B2 Deletion on Wheat Grain Number Per Spike (Grain Setting)

The presence of MN in 50% tetrads suggested unbalanced microspores, which could also affect grain set. Two experiments (CER and glasshouse) were therefore conducted to assess the effect of deleting *Tazip4-B2* on grain set. In these experiments, grain setting analysis was performed on both the CRISPR *Tazip4-B2* mutant and the *ph1b* hexaploid wheat mutant carrying the 59.3 Mb deletion covering *Tazip4-B2*. Spikelet number was recorded, as well as number of grains per spike for the first three spikes from each mutant and their corresponding WTs. The normalized grain number per spike was used to compare genotypes. Both CER and glasshouse experiments confirmed significantly reduced seed set in both *Tazip4-B2* mutants compared to their corresponding WT (*p* < 0.01) (Figure 4a; Table 2). Under CER conditions, the grain number per spike was reduced by 36% in the CRISPR *Tazip4-B2* mutant compared to the WT Fielder, and by 42% in the *ph1b* mutant compared to the Chinese Spring WT. Under glasshouse conditions, the grain number per spike was reduced by 44% in the CRISPR *Tazip4-B2* and 43% in the *ph1b* mutant, compared to their corresponding WTs. There was no significant difference between the CER and glasshouse growth conditions on grain settings for each genotype (Table 2 and Appendix A). Thus, the CRISPR deletion of *TaZIP4-B2* in hexaploid wheat resulted in 56% of meiocytes exhibiting meiotic abnormalities, 50% of tetrads exhibiting micronuclei, and up to 44% reduction in grain set. Similarly, the *ph1b* mutant also exhibited 56% meiocytes with meiotic abnormalities and up to 43% reduction in grain set.

### 3.4. Pollen Contributes to the Tazip4-B2 Effect on Grain Setting

As previously described, on the female side, only one of the 4 megaspores develops into the embryo sac, with the 3 remaining megaspores degenerating following the tetrad stage. This contrasts with the male side, where all four products of meiosis survive to go through pollen development. It is possible that on the female side, some of the unbalanced megaspores are aborted, so that the near 50% reduction in grain set observed in the CRISPR *Tazip4-B2* mutant mostly results from pollination with less viable pollen. An emasculation/pollination experiment was therefore conducted, using the CRISPR *Tazip4-B2* mutant and its corresponding WT. The experiment involved pollinating WT plants with WT or *Tazip4-B2* mutant pollen, or the *Tazip4-B2* mutant with WT pollen. Results showed that the lowest percentage of grain number per spike occurred when WT plants were pollinated with CRISPR *Tazip4-B2* mutant pollen (Figure 4b; Appendix A), and that this grain set was significantly lower than that produced by pollinating WT plants with WT pollen (37.8% difference; *p* < 0.01) (Figure 4b). In contrast, when the *Tazip4-B2* mutant was pollinated with WT pollen, the reduction in grain set was not significantly different to when WT plants were pollinated with WT pollen (Appendix A). These results suggest that most of the reduced grain number in the CRISPR *Tazip4-B2* mutant is due to its being pollinated with less viable pollen, rather than it all being due to impaired female gametogenesis. Thus, meiotic abnormalities associated with *TaZIP4-B2* deletion may have a greater subsequent effect on male gametogenesis than on female gametogenesis.

The maternal and paternal effects of *TaZIP4-B2* on seed embryo development were assessed by germinating the resulting seeds from each of the above pollination experiments. Germination rates from each of the pollination experiments were not significantly different (Figure 4c; Appendix A). Thus, there was no apparent negative effect of the CRISPR *Tazip4-B2* mutation on the germination of seed derived from WT plants pollinated with *Tazip4-B2* mutant pollen, or from *Tazip4-B2* mutants pollinated with WT pollen.

### 3.5. A Pollen Profiling Approach Reveals 50% Tazip4-B2 Mutant Pollen Is Small

Meiotic abnormalities in 56% meiocytes lead to mis-segregation of chromosomes and 50% tetrads with micronuclei. The *Tazip4-B2* mutant has up to a 44% reduction in grain number. The emasculation and pollination experiment suggests that most of this effect is the result of reduced pollen viability. We therefore developed a pollen profiling approach to study the effect of the CRISPR *Tazip4-B2* mutant on wheat pollen size and number, based on approaches previously described by Lamborn et al. [42] and De Storme et al. [43], but modified to allow simultaneous and precise measurement of pollen grain size and pollen number per anther. The modified method was validated using pollen samples from five different wheat varieties, namely Cadenza, Fielder and Paragon (hexaploid), Cappelli and Kronos (tetraploid) and one hexaploid wheat landrace (Chinese Spring). Fully mature anthers were collected from the middle portion of the first ear of each plant, just before opening and pollen shedding, and stored in 70% ethanol. The samples could be stored in ethanol for a long period before analysis, without significant effect on pollen measurement accuracy. Pollen profiles of anther samples in 70% ethanol from the same genotype after different storage periods (of up to one month) are shown in Appendix A. Sonication was used to ensure that all pollen grains were released from anthers, ensuring accurate measurement of pollen number per anther. Pollen size measurements from the six wheat varieties showed that the average pollen size in the hexaploid wheats was 49.0 ± 0.4 µm, (ranging from 48.6 ± 1.2 µm to 49.5 ± 1.1 µm in Chinese Spring and Paragon, respectively), while in the tetraploid wheats it was 44.6 ± 0.2 µm (44.8 ± 1.4 µm and 44.4 ± 1.4 µm in Cappelli and Kronos, respectively) (Table 3 and Appendix A), in keeping with previously reported wheat pollen sizes [44,45]). Pollen profiles of the hexaploid and tetraploid wheat varieties are shown in Figure 5a.

Pollen number per anther varied between different wheat varieties (Figure 5b). The average number of pollen grains per anther was 2709 ± 614 (ranging from 1973 ± 272 in Fielder to 3515 ± 260 in Cappelli) (Table 3). There was no correlation between number of pollen grains and polyploidy level, as there were no significant differences between hexaploid wheats (Cadenza and Paragon) and tetraploid wheats (Kronos and Cappelli) (Appendix A). Nevertheless, pollen numbers were in keeping with those reported in a previous study [46]. The pollen profiling method allowed us to compare pollen grain size distribution and pollen number from three different *Tazip4-B2* mutants with the relevant WT controls. Pollen was collected from full mature anthers (just before opening) for each of the *Tazip4-B2* mutants (CRISPR *Tazip4-B2; ph1b* hexaploid wheat mutant carrying a 59.3 Mb chromosome 5B deletion covering *TaZIP4-B2*; *ph1c* tetraploid mutant carrying a large deletion of chromosome 5B covering *TaZIP4-B2*) and their WTs (*T. aestivum* cv. Chinese Spring; *T. turgidum* subsp. *Durum* cv. Senatore Cappelli [26]; *T. aestivum* cv. Fielder, respectively). Ten to twelve biological replicates for each of the six genotypes were included in the experiment. Pollen grain size and number were measured from five samples of each biological replicate using the Coulter counter Multisizer 4e. In this study, a mean of 10,948 ± 2063 pollen grains were measured from each genotype (average 1121 ± 208 pollen grains per plant) (Table 4). The three *Tazip4-B2* mutants showed a consistent and similar pollen profile comprising of two distinct peaks. The first peak represented pollen grains with grain size distribution similar to WT pollen and the second a group of pollen grains with smaller grain size (Figure 6a). Accordingly, there were significant differences between the mean pollen grain size of each of the mutants *ph1b*, *ph1c* and CRISPR *Tazip4-B2* and their corresponding WTs (*p* < 0.01) (Table 4). More than 48% of pollen grains in the CRISPR *Tazip4-B2* hexaploid mutant samples were smaller in size (≤42 µm). A similar percentage of small pollen grains (47%) was found in the *ph1b* hexaploid mutant samples. However, small pollen grains (≤38 µm) were found in a lower percentage (34%) in the *ph1c* tetraploid mutant samples (Figure 6b). The mean pollen number per anther ranged from 2317 ± 333 to 3713 ± 497 in the CRISPR *Tazip4-B2* and *ph1c* mutants, respectively (Figure 6c). However, no significant differences were observed between any of the *Tazip4-B2* mutants and their WT controls (Table 4). Detailed datasets of pollen size, pollen number per anther and percentage of small pollen grains for each *TaZIP4-B2* mutant and its respective wild type can be found in Appendix A.

Viability of pollen from the CRISPR *Tazip4-B2*, and *ph1b* hexaploid mutants, as well as the *ph1c* tetraploid mutant, was assessed using Alexander staining. More than 3000 pollen grains were scored for each genotype (from three biological replicates) after Alexander staining and image acquisition. Pollen coloured dark magenta after treatment with Alexander stain was considered viable, whereas light blue-green stained pollen was considered unviable (Figure 7a). Analysis revealed similar percentages of unviable pollen grains in all *Tazip4-B2* mutants (Figure 7c), with 28% in the CRISPR-*Tazip4-B2* mutant, 25.8% in the *ph1b* mutant and 22.8% in the *ph1c* mutant pollen being unviable (Appendix A). In all cases, the level of unviable pollen grains in the mutants was significantly higher than that in the WTs (*p* < 0.01), which did not exceed 3.3% on average. Developmental pollen stages were also assessed in the *TaZIP4-B2* mutants. Pollen grains from fully mature anthers from the Fielder mutant CRISPR *Tazip4-B2* and the WT Fielder were stained selectively for DNA using Feulgen stain. Results from the WT showed normal trinucleate pollen grains, whereas about half of the pollen grains in the mutant were immature and/or abnormal (Figure 7b). Thus, the pollen profiling analysis revealed that around half the pollen from the CRISPR *Tazip4-B2* and *ph1b* hexaploid wheat mutants had similar pollen profiles, with around half the pollen grains being abnormally small. The Alexander and Feulgen staining methods provided further information revealing that the small pollen grains in the CRISPR *Tazip4-B2* and *ph1b* mutants were a mixture of both immature (unfunctional) and unviable pollen grains.

## 4. Discussion and Conclusions

Polyploid literature highlights the requirement for a newly developed polyploid to retain production of balanced gametes to preserve fertility, through both meiotic and genomic adaptations [1,2,3,4]. However, surprisingly few of these polyploid meiotic adaptions have been characterised. Recent studies have assessed the effect of natural gene variation on the meiotic stabilisation process. For instance, at least eight different autotetraploid Arabidopsis meiotic genes have been implicated in meiotic stabilisation [47], with specific alleles in one of these genes, *ASY3*, being highlighted in two further studies [48,49]. Allotetraploid Brassica studies have also reported a number of loci exhibiting natural variation in their ability to affect homoeologous pairing and crossover [50,51,52].

In the present study, we have assessed the stabilising effects of the meiotic gene *TaZIP4-B2*, which arose on chromosome 5B during wheat polyploidisation, through duplication from chromosome 3B [18,19,20]. Results indicate that the deletion of the duplicated *TaZIP4-B2* copy (through CRISPR deletion of *Tazip4-B2*) results in 56% of meiocytes exhibiting meiotic abnormalities at metaphase I (Figure 2 and Figure 3a,b); chromosome mis-segregation at anaphase I (Figure 3c–f); 50% of tetrads possessing micronuclei (Figure 3g–j); and lastly, 48% of pollen grains being small (a mixture of immature and unviable) (Figure 6 and Figure 7). A similar level of disruption is also observed in a hexaploid mutant (*ph1b*) carrying a 59.3 Mb deletion encompassing *TaZIP4-B2*, with 56% of meiocytes exhibiting meiotic abnormalities (Figure 2) and 47% of pollen grains being small (Figure 6). Results suggest a direct correlation between meiotic abnormalities observed at metaphase I and pollen fertility. Importantly, there is also up to a 44% reduction in grain set in the CRISPR *Tazip4-B2* mutant (43% reduction in the *ph1b* mutant) (Figure 4a). A considerable part of this reduction in grain set is likely to be due to pollination with immature/unviable pollen (Figure 4b), rather than being mainly due to disruption in female gametogenesis. Pollen deposition and pollen grain size can have an effect on pollen competition for the ovule [53,54]. However, it is still unclear how, within the 50:50 mixture of WT and immature/unviable pollen, WT pollen does not compete more effectively during pollination.

Development of in situ approaches are required to explain the observed apparently lesser effect of *Tazip4-B2* on subsequent female gametogenesis. It is likely that, as in male meiocytes, meiotic abnormalities are exhibited in female meiocytes at metaphase I and MN during the tetrad stage. Study of this tetrad stage, when only a single megaspore remains [41], will identify any preferential abortion of megaspores with unbalanced chromosome numbers due to disrupted meiotic pairing and crossover. However, whatever the importance of *Tazip4-B2* for female gametogenesis, the presence of *TaZIP4-B2* was still required to ensure nearly half the hexaploid wheat grain set, confirming the great importance and impact of the *ZIP4* duplication event on the fertility of this major global polyploid crop.

In diploids, mutation of a single copy meiotic gene is often associated with meiotic abnormality, reduced fertility or sterility. However, in polyploids, this is generally not the case, particularly in a hexaploid such as bread wheat. For example, hexaploid wheat mutants carrying single copy deletions covering the major meiotic genes *DMC1*, *ASY1*, *FIG1*, *SPOII* and *MLH1*, largely exhibit a wild type phenotype, when configurations are scored at metaphase I [15]. Similarly, in hexaploid wheat, all three copies of *SPOII.2* require mutation to produce sterility [55], while in tetraploid wheat, both copies of *MSH4* and *MSH5* require mutation to produce a meiotic phenotype [56]. However, unusually, hexaploid wheat mutants with deletions covering *TaZIP4-B2* in the long arm of chromosome 5B, did exhibit meiotic abnormalities. This occurrence of a meiotic pairing and crossover phenotype leading to reduced fertility with loss of a single copy of *TaZIP4-B2* is surprising, given the presence of three other *ZIP4* copies in the hexaploid wheat genome.

Recently, a study has characterised *Ph2* in wheat as being *MSH7-3D*, which, in common with *TaZIP4-B2*, suppresses homoeologous crossover [57]. However, *TaZIP4-B2* also has a major effect on chromosome pairing, by promoting homologous pairing. Interestingly, in contrast to the *Tazip4-B2* mutants reported here, *msh7-3D* mutants do not exhibit a significant reduction in grain number and only a 20% reduction in pollen viability. This suggests that when considering preservation of pollen viability and grain number, the ability of *TaZIP4-B2* to promote homologous pairing is more important than its ability to suppress homoeologous crossover. In budding yeast, *ZIP4* reduces premature separation of sister chromatids at anaphase I [58]. In the present study, *Tazip4-B2* deletion mutants exhibit extensive premature sister chromatid separation (Figure 3). A previously described *Tazip4-B2* TILLING mutant (termed Cad1691), carrying a single amino acid change affecting both homologous and homoeologous crossover, but not homologous pairing [18] could now be exploited to confirm the link between promotion of homologous pairing and reduction of premature sister chromatid separation. Fertility scoring of this same *Tazip4-B2* TILLING mutant could also confirm whether the *TaZIP4-B2* homologous pairing effect is indeed more important for preservation of fertility than its effect on homoeologous crossover suppression.

In hexaploid wheat, the duplicated *TaZIP4-B2* copy promotes homologous pairing, synapsis and crossover, and suppresses homoeologous crossover, so preserving pollen viability and grain set. In Arabidopsis [21] and rice [22], *ZIP4* is required for 85% of homologous crossovers during meiosis, but studies have not shown a role for *ZIP4* in pairing and/or synapsis. In Sordaria [59] and budding yeast [58], *ZIP4* is required for pairing and synapsis, as well as homologous crossover; in Sordaria, *ZIP4* foci form ‘pairing bridges’ between chromosomes. The hexaploid wheat *TaZIP4-B2* phenotypes most likely result from a reduction in the normal functions of group 3 *ZIP4s*, as a consequence of the TPR divergence within *TaZIP4-B2* from that within *TaZIP4-B1* (Figure 1b,c). The early and 3-fold increased expression of *TaZIP4-B2* compared to the group 3 *ZIP4s*, is also likely to ensure that it competes with them for loading onto meiotic chromosomes [18]. The present study reveals that up to half the wheat *ZIP4* protein is composed of TPRs (Figure 1b,c). The presence of TPRs in other proteins has been shown to enable these proteins to form alpha solenoid helix structures capable of assembling protein complexes [34,35]. Recent studies in budding yeast have revealed that *ZIP4* interacts with axis and crossover proteins, which may provide the basis of its effects in wheat [60]. Previous studies have suggested that the *ph1b* deletion effect on homoeologous crossover in wheat is linked to the improved ability of the meiotic crossover protein MLH1 to process crossovers [38], while the *ph1b* deletion effect on chromosome pairing reported by Roberts et al. [15] is linked to the chromosome axis protein, ASY1 [61]. Recent studies in budding yeast have revealed that *ZIP4* is connected to MLH1 through the binding to MER3, to HOP1 (ASY1) through the binding of another chromosome axis protein RED1 (ASY3), and to synapsis proteins through ZIP2 [60].

The promotion of homologous pairing by *TaZIP4-B2* is likely to be related to the elongation of the chromosome axis on entry into meiosis [62], and the association of *ZIP4* foci on these axes forming ‘pairing bridges’ between homologues [59]. If the extent of homologue elongation is different, then homologue alignment and pairing is delayed during early meiosis [63]. REC8, a cohesion protein, is required for correct meiotic chromosome conformation, and chromosome axis elongation via assembly of ASY1 [64,65]. *ZIP4* locates at the end of REC8-associated chromatin regions [22]. Thus, the simplest explanation for the ability of *TaZIP4-B2* to promote homologous pairing is that *TaZIP4-B2* reduces homologue elongation, ensuring more similar conformations and allowing rapid association of *ZIP4* foci, so reducing the chance of homoeologous pairing which occurs later in meiosis [28,38,62].

The suppression of homoeologous crossover by *TaZIP4-B2* is likely to result from the interplay between this chromosome 5B copy and the three group 3 chromosome *ZIP4* copies. The group 3 *ZIP4s* are likely to process 85% of homologous crossovers as in other species [21,22]. They are also likely to process homoeologous crossover activity, given the level of crossover observed in wheat haploids lacking *TaZIP4-B2* [66]. In contrast, although the diverged *TaZIP4-B2* copy has some homologous crossover activity, it does not possess any homoeologous crossover activity [19]. Sordaria studies reveal that initial chromosome interactions involve *ZIP4* foci on homologous chromosomes [59]. Thus, the presence of *TaZIP4-B2* with wheat group 3 *ZIP4s* in chromosome foci which assemble crossover proteins including MLH1, means that only homologous crossovers are successfully processed rather than homoeologous crossovers [14,19,38].

Deletion of *TaZIP4-B2* reduces homologous crossover, contributing to an increase in meiotic abnormalities at metaphase I (Figure 2) [19]. This suggests that the presence of *TaZIP4-B2* increases homologous crossover, indicating that the *ZIP4* effect on homologous crossover may be dosage dependent. This contrasts with other meiotic genes analysed in polyploid Brassica and wheat, where loss of a single gene copy does not reduce overall homologous crossover, with homologous crossover only being affected if all gene copies are deleted [56,67]. This also highlights that the significant reduction in polyploid fertility with the loss of one of the gene copies (in this case, *TaZIP4-B2*), may be unusual for meiotic genes in polyploids. Thus, *ZIP4* was an effective target for divergence on polyploidisation, as its homologous crossover activity appears to be dosage dependent.

Given the importance of the *TaZIP4-B2* function for preserving grain number in wheat, future studies will need to confirm that the phenotype of *TaZIP4-B2* results from a reduction in the function activities possessed by the group 3 *ZIP4s*. Such studies will also need to confirm whether different *TaZIP4 B2* alleles exhibit variable phenotypes sensitive to temperature change. Natural variation in meiotic phenotypes has been reported for some meiotic genes in other polyploids [47,48,49,50,51,52]. The approach for analysing pollen presented in this study can be used in future *TaZIP4-B2* studies to screen landrace diversity mapping populations [68], carrying different *TaZIP4-B2* alleles for variable phenotypes, with variable sensitivity to temperature. This approach is high throughput and sensitive, with the capability to screen 1000s of pollen grains rapidly. Thus, the approach can be used for forward and reverse meiotic genetic screening. The approach has already been successfully used in a forward genetics screen to identify temperature sensitive mutants in hexaploid wheat (unpublished). In the present study, the technique was used to analyse pollen derived from both tetraploid and hexaploid meiotic mutants, revealing the presence of small pollen (Figure 6a,b). The recent availability of multiple sequenced wheat genomes has allowed the initial identification of haplotype blocks [68], revealing different *TaZIP4-B2* haplotypes. This information, combined with the availability of landrace diversity mapping populations [69] and the pollen technique, can be used to rapidly identify any potential natural phenotype variation correlating with a specific *TaZIP4-B2* haplotype, as well as to explore the stability of such phenotypes under variable temperatures. This will be important for studies exploring the effects of temperature increases on wheat yields within the context of global climate change.

## Figures and Tables

**Figure 1 biology-10-00290-f001:**
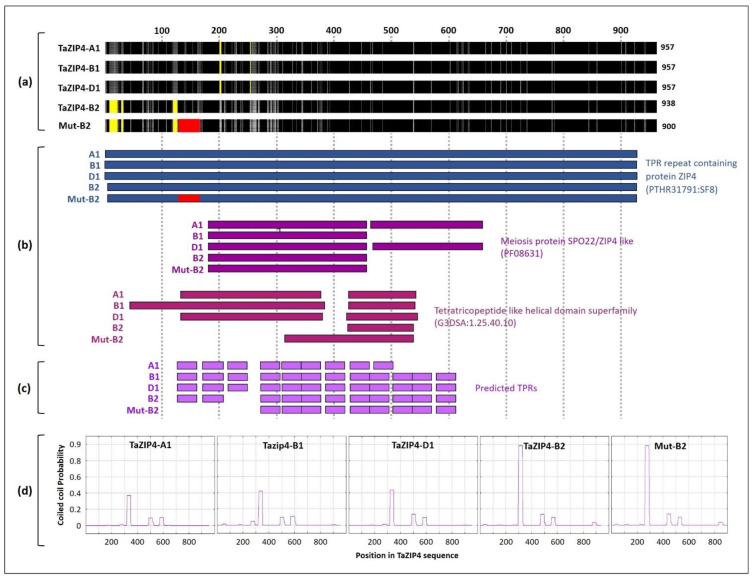
Comparison of the TaZIP4 homoeologous proteins. (**a**) Multiple amino acid sequence alignment of the TaZIP4 homoeologous proteins using ClustalX software (version 2.0; [31,32]). Regions with identical amino acid sequences across the four proteins are in black. Grey colour refers to sequences with similar amino acid properties and light grey refers to sequences with different amino acid properties. Yellow regions indicate gaps in sequence alignment. Mut-B2 refers to *Tazip4-B2* in the CRISPR mutant. Red region shows the 38-amino acids segment deleted from the protein of the CRISPR *Tazip4-B2* mutant. (**b**) Predicted functional domains in the TaZIP4 proteins using the online InterPro software (version 82.0; [33]). (**c**) Predicted Tetratricopeptide Repeats (TPRs) in the TaZIP4 proteins using the online TPRpred program (version 11.0; [36]. (**d**) Predicted coiled coil domains in the TaZIP4 proteins using the online MARCOIL programme [36,37].

**Figure 2 biology-10-00290-f002:**
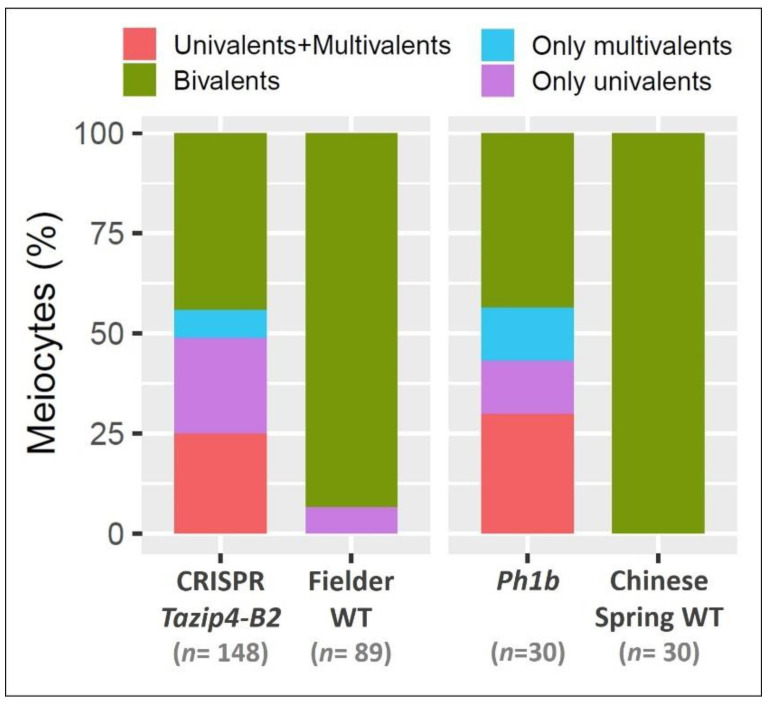
The percentage of meiocytes with meiotic abnormalities from the CRISPR *Tazip4-B2* and *ph1b* mutants, and their wild types. Data used to produce this figure is taken from Rey et al. [19] and Martín et al. [38]. *n* refers to the number of scored meiocytes.

**Figure 3 biology-10-00290-f003:**
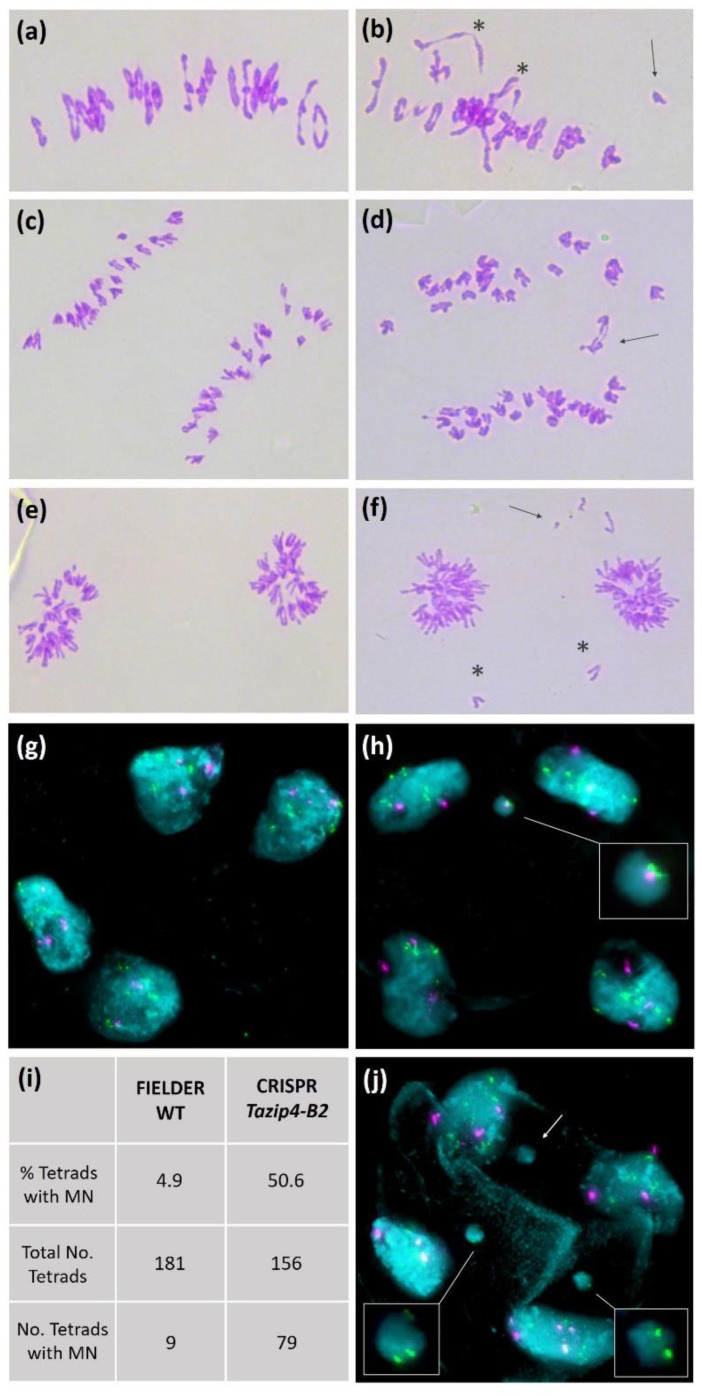
Meiosis in wild type (WT) Fielder (**a**,**c**,**e**,**g**) and the CRISPR *Tazip4-B2* Fielder mutant (**b**,**d**,**f**,**h**,**j**). (**a**) Metaphase I in WT Fielder showing 19 ring bivalents and 3 rod bivalents. (**b**) Metaphase I in CRISPR *Tazip4-B2* mutant with the presence of multivalents (asterisk) and univalent (arrow). (**c**) Anaphase I in WT displaying equal separation of homologous chromosomes to both poles. (**d**) Anaphase I in CRISPR *Tazip4-B2* mutant showing lagging chromosomes. (**e**) Late anaphase I in WT. (**f**) Late anaphase I in CRISPR *Tazip4-B2* mutant showing some chromosome fragments in the periphery of equatorial plate (arrow) and chromosome mis-division in the equatorial plate (asterisk) which will not be included in any of the dyads. (**g**,**h**) Tetrads shown in cyan, with repetitive probe 4P6 (in green) and pTa71 (in magenta). (**g**) Tetrad from WT showing 4 normal microspores. (**h**) Tetrad in CRISPR *Tazip4-B2* mutant showing 1 micronucleus (MN) displaying 4P6 and pTa71 signals. (**j**) Tetrad in CRISPR *Tazip4-B2* mutant showing 3 micronuclei, 2 of them presenting 4P6 signals. (**i**) Close to 5% of the tetrads showing MN in WT, while 50% of tetrads in the CRISPR *Tazip4-B2* mutant possessing 1, 2 or 3 MN.

**Figure 4 biology-10-00290-f004:**
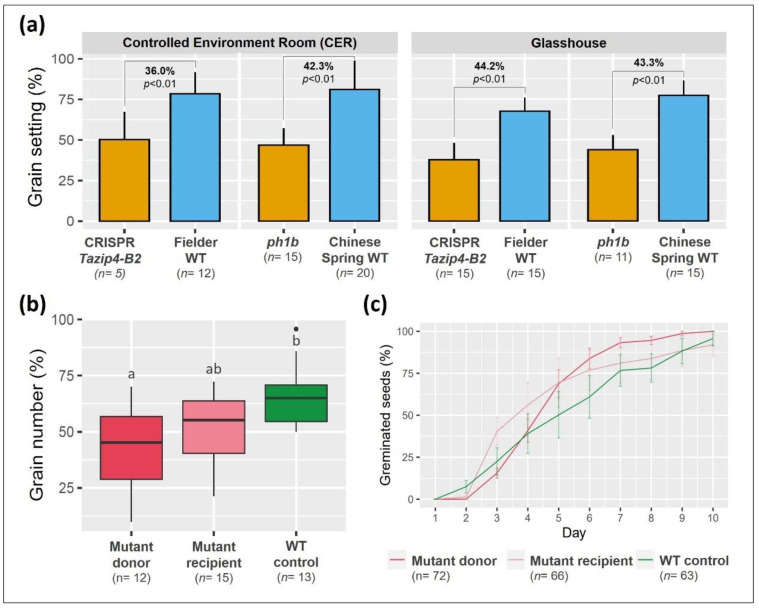
The effect of *TaZIP4-B2* on grain setting. (**a**) Grain number per spike in the two *Tazip4-B2* mutants and their WT controls under CER and glasshouse growth conditions. Percentages indicate the difference in grain setting between each mutant and its WT. *n* refers to the number of biological replicates. (**b**) Normalised grain number per spike in the three treatments of the emasculation/pollination experiment. Treatments with the same letter are not significantly different. *n* refers to the number of emasculated/pollenated spikes. (**c**) Seed germination rates resulting from different pollen donor and pollen recipient genotypes. *n* refers to the number of seeds included in the seed germination experiment.

**Figure 5 biology-10-00290-f005:**
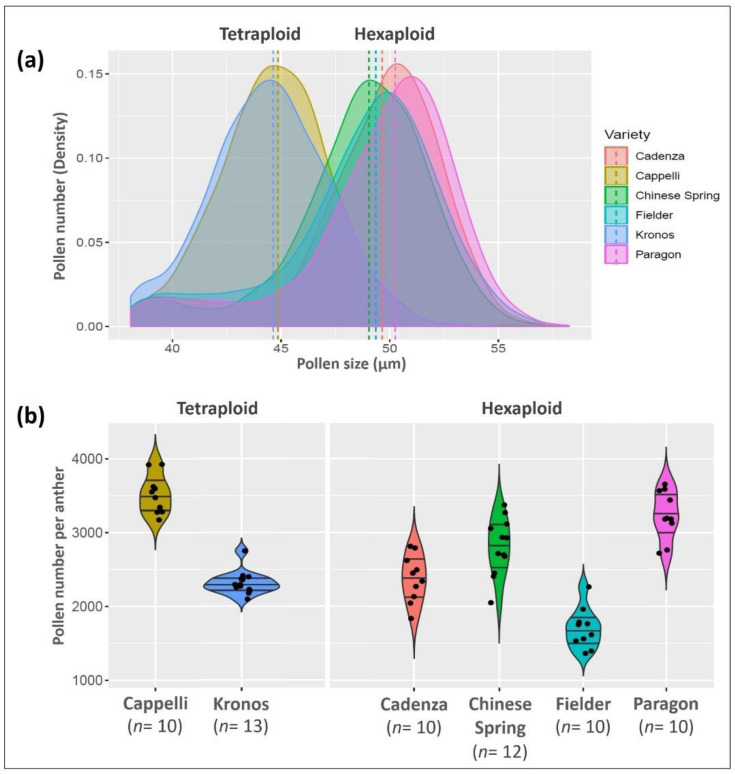
Pollen size and number per anther of some hexaploid and tetraploid wheats. (**a**) Density plot of the differential pollen size distribution data collected by coulter counter (Multisizer 4e) for four hexaploid wheats (Chinese Spring, Cadenza, Fielder and Paragon) and two tetraploid wheats (Cappelli and Kronos). Dotted lines indicate the median pollen grain size for each genotype. (**b**) Pollen number per anther for the six mentioned hexaploid and tetraploid wheat varieties. *n* is number of plants (biological replicates).

**Figure 6 biology-10-00290-f006:**
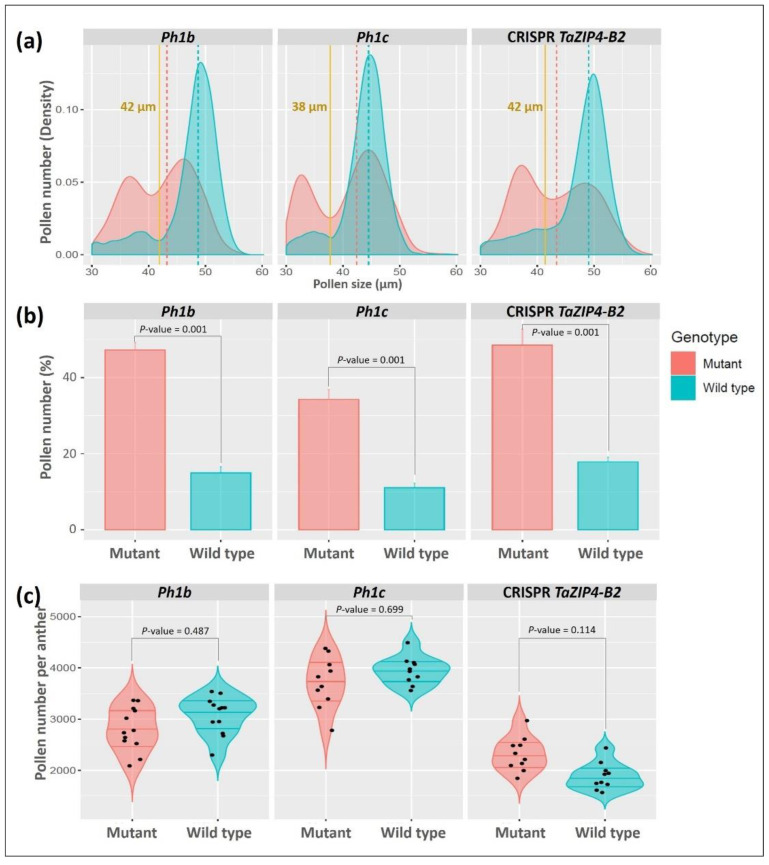
Pollen profiles of the three *Tazip4-B2* mutants. (**a**) Density plot of the differential pollen size distribution data collected by coulter counter (Multisizer 4e) showing two distinguished peaks in all *Tazip4-B2* mutants compared to their corresponding wild types. Dotted lines indicate the median pollen grain size for each genotype. Yellow lines indicate the borderline between normal and small pollen for each genotype group. (**b**) Percentages of small pollen grains for each genotype (mutants and wild types). Pollen grain is considered small when it is ≤42 µm and ≤38 µm for hexaploid and tetraploid wheat pollen size respectively. (**c**) Comparison of number of pollen grains per anther between each *Tazip4-B2* mutant and its wild type. No significant difference in pollen number per anther was found between any of the mutants and its wild type.

**Figure 7 biology-10-00290-f007:**
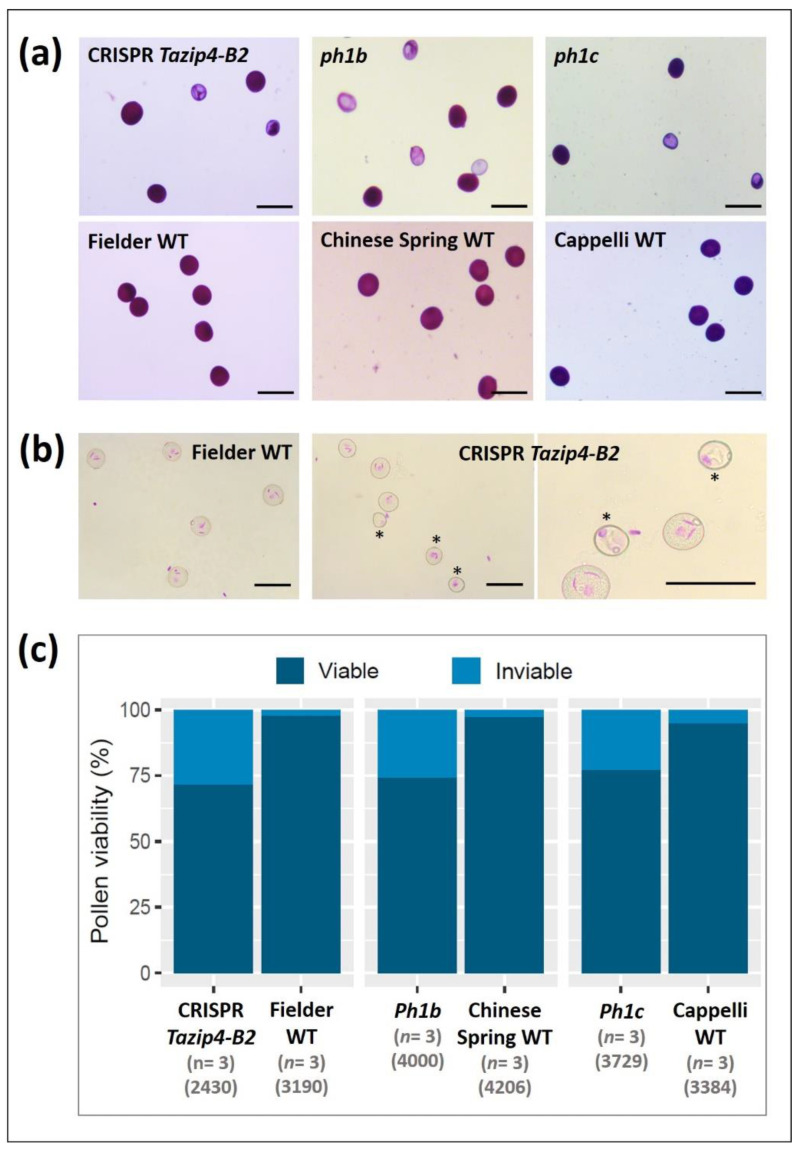
Pollen viability of the *Tazip4-B2* mutants. (**a**) Pollen with magenta colour after staining with Alexander stain was considered viable, whereas blue-green pollen was considered unviable. Bars equal 100 µm in length. (**b**) Feulgen staining of pollens from anthers at anthesis in the CRISPR *Tazip4-B2* mutant and its wild type (cv. Fielder) shows normal trinucleate pollen grains in the wild type, whilst almost half of pollens were immature and/or abnormal in the mutant. Immature and abnormal pollen grains are indicated by an asterisk. Bars equal 100 µm in length. (**c**) Percentages of viable and unviable pollens according to the Alexander staining method for the three *Tazip4-B2* mutants and their wild types. *n* refers to the number of biological replicates. Numbers between brackets refer to the total number of scored pollen grains for each genotype.

**Table 1 biology-10-00290-t001:** Summary of the meiotic scores for the two *Tazip4-B2* mutants and their corresponding wild types.

Genotype	Univalents	Multivalents	Rod Bivalents	Ring Bivalents	Defective Meiocytes * (%)	Reference
**Fielder WT ****	0.16 ± 0.07	0 ± 0	1.37 ± 0.13	19.52 ± 0.14	6.70	[19]
**CRISPR *Tazip4-B2***	1.16 ± 0.12	0.39 ± 0.05	4.93 ± 0.15	14.84 ± 0.19	56.00	
**Chinese Spring**	0 ± 0	0 ± 0	1 ± 0.20	20 ± 0.20	0.00	[38]
***ph1b***	0.80 ± 0.19	0.53 ± 0.12	4.73 ± 0.26	14.83 ± 0.33	56.60	

* Meiocytes with meiotic aberrations (univalents and multivalents) thus have incorrect chromosome pairing. ** This is a CRISPR transgenic Fielder without *TaZIP4-B2* knockout.

**Table 2 biology-10-00290-t002:** Mean Normalized grain number per spike for the two *Tazip4-B2* mutants and their corresponding wild types under CER and glasshouse growth conditions. N is the number of biological replicates per genotype. Mean values with standard deviation are shown. Treatments with the same letter are not significantly different.

Genotype	Controlled Environment Room (CER)		Glasshouse	
	N	Normalized Grain Number Per Spike	N	Normalized Grain Number Per Spike
CRISPR Tazip4-B2	5	50.2 ± 16.7 ^b,c,d^	15	37.7 ± 1.0 ^d,e^
cv. Fielder WT	12	78.5 ± 12.9 ^a^	15	67.6 ± 07.9 ^a,b^
ph1b	15	46.7 ± 1.0 ^c,d^	11	43.8 ± 08.6 ^c,d^
cv. Chinese Spring WT	20	80.9 ± 17.3 ^a^	15	77.3 ± 08.8 ^a^

**Table 3 biology-10-00290-t003:** Pollen number and pollen grain size for some hexaploid and tetraploid wheat varieties. Mean and median values with standard deviations are shown. Treatments with the same letter are not significantly different.

Polyploidy	Variety	N	Pollen Size		Pollen Number Per Anther	
			Mean	Median	Mean	Median
Hexaploid	Cadenza	10	49.05 ± 1.28 ^a^	49.65 ± 1.11 ^a^	2380 ± 320 ^b^	2407 ± 348 ^b^
Hexaploid	Chinese Spring	12	48.58 ± 1.17 ^a^	49.05 ± 1.15 ^a^	2807 ± 384 ^c^	2840 ± 408 ^c^
Hexaploid	Fielder	10	48.67 ± 0.98 ^a^	49.36 ± 1.19 ^a^	1973 ± 272 ^a^	1686 ± 250 ^a^
Hexaploid	Paragon	10	49.51 ± 1.09 ^a^	50.25 ± 1.22 ^a^	3318 ± 236 ^d^	3293 ± 339 ^d^
Tetraploid	Cappelli	10	44.77 ± 1.40 ^b^	44.85 ± 1.48 ^b^	3515 ± 260 ^d^	3480 ± 277 ^d^
Tetraploid	Kronos	13	44.44 ± 1.39 ^b^	44.63 ± 1.52 ^b^	2260 ± 110 ^b^	2373 ± 256 ^b^
All hexaploid wheat varieties		4	48.95 ± 0.43	49.58 ± 0.51	2533 ± 657	2557 ± 684
All tetraploid wheat varieties		2	44.61 ± 0.23	44.74 ± 0.15	2915 ± 849	2926 ± 783

**Table 4 biology-10-00290-t004:** Pollen number and pollen grain size for the three *Tazip4-B2* mutants and their corresponding wild types. Mean and median values with standard deviations are shown. Treatments with the same letter are not significantly different.

Genotype	Total Number of Measured Pollen Grains	N	Pollen Grain Size (µm)		Pollen Number Per Anther		Small Pollen Grains (%)	
			Mean	Median	Mean	Median	Mean	Median
*ph1b*	10100	12	42.3 ± 1.3 ^a,b^	43.2 ± 1.7 ^a^	2806 ± 426 ^a^	2817 ± 430 ^a^	47.2 ± 6.9 ^a^	45.1 ± 8.3 ^a^
cv. Chinese Spring	11072	12	47.4 ± 1.3 ^c^	48.7 ± 1.5 ^b^	3076 ± 370 ^a^	3082 ± 364 ^a^	15.0 ± 5.6 ^b^	13.2 ± 2.5 ^b^
*ph1c*	11139	10	40.9 ± 1.9 ^b^	42.4 ± 2.6 ^a^	3713 ± 497 ^c^	3681 ± 593 ^c^	34.3 ± 8.2 ^c^	34.4 ± 8.3 ^c^
cv. Cappelli	11840	10	43.7 ± 1.5 ^a^	44.5 ± 1.5 ^a^	3947 ± 270 ^c^	3950 ± 248 ^c^	11.1 ± 3.8 ^b^	10.7 ± 4.0 ^b^
CRISPR *Tazip4-B2*	13904	10	43.1 ± 2.1 ^a^	43.4 ± 3 ^a^	2317 ± 333 ^b^	2354 ± 379 ^a,b^	48.5 ± 13.4 ^a^	48.6 ± 13.4 ^a^
cv. Fielder	11313	10	47.3 ± 1.0 ^c^	49.1 ± 1.2 ^b^	1886 ± 265 ^b^	1875 ± 231 ^b^	17.9 ± 3.9 ^b^	19.1 ± 6.3 ^b^

## Data Availability

Datasets supporting the findings of this study are available in the Appendix A of this article and from the corresponding author upon reasonable request.

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
