# Peer review of "A Duplicated Copy of the Meiotic Gene ZIP4 Preserves up to 50% Pollen Viability and Grain Number in Polyploid Wheat"

_biology, 2021, doi:10.3390/biology10040290_

Round 1

Reviewer 1 Report

Compared to an earlier version of the manuscript, the authors have addressed most of the concerns/doubts raised in this new and improved version. Below is copied in the general summary from the initial version.

Alabdullah et al dissect the role of TaZIP4-B2 on wheat fertility employing three different mutant lines (ph1b, CRISPR Tazip4-B2 and ph1c) and their respective wildtype backgrounds. The authors and others showed earlier that ph1 (primarily TaZIP4-B2) promotes meiotic pairing, synapsis and crossover formation involving homologous chromosomes while it suppresses formation of crossover between homeologous chromosomes. In the current study, the impact of ph1/Tazip4-B2 on fertility employing various techniques (pollen profiling, pollen viability tests, grain number counts, seed germination rates, pollination experiments or meiotic chromosome studies) was studied showing that roughly 50% of male meiocytes and their subsequent products (tetrads and pollen grains) are defective. Interestingly, the reduction in seed set found in ph1/Tazip4-B2 seems to be primarily a consequence of male meiotic defects rather than based on reduced female fertility as suggested by performed reciprocal crosses/pollination experiments. Moreover, using various in silico tools and approaches the authors compare and discuss difference among the ZIP4 homeologous proteins. As discussed by the authors, further experiments in future could shed light on the role of ph1/TaZIP4-B2 for female reproduction, i.e. in particular female meiosis and female gamete development.

The rationale of the work is clear. Represented data are solid and clear as well as the manuscript is easy to follow and nicely written. The study including its methodology is not necessarily ground-breaking but it concerns an important crop and in particular the role of ph1/TaZIP4-B2 for wheat polyploidization/reproduction. Therefore, it will be of interest for breeders/researchers working on wheat or working on other species in the context of meiotic stability and polyploidization. It it will most certainly also stimulate further research on the role of ph1/TaZIP4-B2 for female fertility.

Author Response

We thank this referee for their comments.   

We have included a further comment in the abstract to highlight the potential for future research on TaZIP4-B2 and its role in female fertility.

Reviewer 2 Report

This is an excellent manuscript building on previous work by this group detailing that the extra ZIP4 copy on chromosome 5B is required for meiotic chromosome stabilisation in allopolyploid wheat. A thorough analysis of metaphase I chromosomes, pollen viability and tetrad analysis shows that the extra ZIP4 copy is essential for normal fertility and not just required to bias recombination towards the homologous chromosomes. There is also an interesting difference between male and female fertility in the 5B (B2) mutant background. The data analysis is appropriate and the figures are well presented.

Minor points:

Line 200. Shouldn’t Tpr be TPR?

I would find it easier to follow if the ZIP4 copies were referred to by chromosome designation such as ZIP4 3A, 3B, 3D and 5B rather than B1 and B2, but that is just a preference. You do actually use this system of labelling on line 317, but it is not clear to me why?

Line 279. A pedantic point, but when referring to cytological crossovers at metaphase I, they should be called chiasmata.

Author Response

We thank the reviewer for their comments. 

The referee comments that it may be easier using the gene nomenclature ZIP4 3B and ZIP4 5B rather than ZIP4-B1 and ZIP4-B2. However the latter terms are those recommended by the international wheat gene nomenclature group, so we have opted to use their terms, rather than the terms ZIP4 3A, ZIP4 3B, ZIP4 3D and ZIP4 5B which we normally use. 

We have changed tpr to TPR (line 253),  ZIP4 5B to ZIP4-B2 (line 397) and added chiasmata (lines 360 and 365) in that paragraph.

Reviewer 3 Report

The study by Alabdullah et al., investigated the effect of a Tazip4-b2 deletion (ph1b) and CRISPR-derived mutants on meiotic abnormalities, pollen viability, number and size as well as seed-set relative to their respective wild-types. Findings indicate that TaZIP4-B2 loss-of-function disrupts chromosome pairing and synapsis as assessed cytologically at metaphase I by an increase in the formation of univalents and multivalents. More in-depth cytological characterisation of only the CRISPR-derived line relative to its wild-type demonstrated an increase in chromosome mis-division and formation of micronuclei. The authors argue that such meiotic abnormalities associate with reduced pollen size and seed-set. Reciprocal crosses demonstrated that the reduced seed-sed is predominantly a consequence of reduced pollen viability as opposed to a non-functional female gametophyte. The data presented in the manuscript used to draw these conclusions is robust and analytical methodology sound. The manuscript is well-articulated and easy to read. However, in terms of consistency and thoroughness, the manuscript could benefit from either the inclusion of additional cytological data demonstrating that ph1b also increases the incidence of micronuclei, or at least point the reader to any relevant reference that reports such a phenomena for ph1b or other similar classic Ph1 mutation. This would clarify that the observed increased incidence of micronuclei is not a characteristic restricted to the CRISPR-mutant alone, and is indeed reflective of what is also disrupted in ph1b. From my interpretation the reference made to Morrison et al., 1953, shows the more general nature of micronuclei formation from univalent mis-segregation in monosomics. Is the difference in univalent formation between ph1b (0.8) and CRISPR Tazip-B2 (1.16) also reflected in the incidence of micronuclei? Providing clarity here would be beneficial.

Minor points:

Figure 1 is out of sequence with the other Figures

Figure 2 is out of sequence with the other Figures

Author Response

We thank the referee for their comments. 

The referee asks for a reference for a classical Ph1 mutant exhibiting micronuclei. We had already included a reference by Giorgi 1983 in our reference list. This paper described micronuclei associated with the tetraploid Ph1 deletion mutant. We have now included a reference to this observation in the text.

The referee queries the difference between 0.8 univalents in the ph1b deletion mutant and 1.16 univalents in the CRISPR Tazip4-B2 mutant. Our view is that these univalents result from incorrect pairing and failure to crossover. It is some 40 years since the ph1b mutant was first generated. We believe that some incorrect pairing during this period has led to chromosome translocations, so that instead of these chromosomes being univalents, they are now from multivalents at metaphase I in the ph1b mutant. As commented in the text, this would explain the apparent slightly higher occurrence of multivalents and lower univalents in the ph1b mutant compared to the newly generated CRISPR Tazip4-B2 mutant. Of course, the mis-segregation of both univalents and multivalents could also contribute  to the occurrence of micronuclei. 

We have corrected the out of sequence Figures.